# CHiLL: Zero-shot Custom Interpretable Feature Extraction from Clinical Notes with Large Language Models

**Denis Jered McInerney**
Northeastern University
mcinerney.de@northeastern.edu

**Geoffrey Young**
Brigham and Women's Hospital
gsyoung@bwh.harvard.edu

**Jan-Willem van de Meent**
University of Amsterdam
j.w.vandemeent@uva.nl

**Byron C. Wallace**
Northeastern University
b.wallace@northeastern.edu

## Abstract

We propose **CHiLL (Crafting High-Level Latents)**, an approach for *natural-language specification of features for linear models.* CHiLL prompts LLMs with expert-crafted queries to generate interpretable features from health records. The resulting noisy labels are then used to train a simple linear classifier. Generating features based on queries to an LLM can empower physicians to use their domain expertise to craft features that are clinically meaningful for a downstream task of interest, without having to manually extract these from raw EHR. We are motivated by a real-world risk prediction task, but as a reproducible proxy, we use MIMIC-III and MIMIC-CXR data and standard predictive tasks (e.g., 30-day readmission) to evaluate this approach. We find that linear models using automatically extracted features are comparably performant to models using reference features, and provide greater interpretability than linear models using "Bag-of-Words" features. We verify that learned feature weights align well with clinical expectations.

## 1 Introduction

LLMs have greatly advanced *few-* and *zero-shot* capabilities in NLP, reducing the need for annotation. This is especially exciting for the medical domain, in which supervision is often scant and expensive. However, given the high-stakes nature of clinical work and the challenges associated with developing models (e.g., long-tail data distributions, weakly informative supervision), predictions can rarely be trusted blindly. Clinicians therefore tend to favor simple models with interpretable predictors over opaque LLMs that rely on dense learned representations. Risk prediction tools are often linear models with handcrafted features. Such models have the advantage of associating features with weights; these can be inspected to ensure clinical tenability and may avoid undesired fragilities of large neural models. However, a downside to relying on inter-

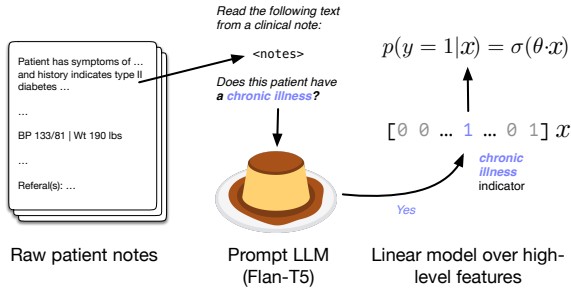

Figure 1: We propose to allow domain experts to specify high-level features for simple linear predictive models in natural language, and then extract these *zero-shot* (without supervision) using large language models.

pretable features is that one often has to manually *extract* them from patient records. This can be particularly difficult when features are high-level and must be inferred from unstructured (free-text) fields within EHR.

In this work we investigate the potential of zero-shot extraction *for definition and abstraction of interpretable features from unstructured EHR to use as inputs for simple (linear) models* (Figure 1). Recent work (Agrawal et al., 2022) has shown that LLMs are capable *zero-shot* extractors of clinical information, i.e., they can extract structured information from records without explicitly being trained to do so. These can be high-level features specified in natural language, which allows practitioners to define features that they hypothesize may be relevant to a downstream task. Such features, even if noisy, are likely to be more interpretable than low-level features such as Bag-of-Words (BoW) representations. The motivation of this work is to evaluate the viability of training small linear models over high-level features automatically extracted via LLMs—without explicit supervision—from unstructured data in EHR.

A secondary concern that this paper seeks to address is any reliance on closed-source LLMs for healthcare, which is undesirable for a number of

reasons. First, such models are inherently opaque, precluding interpretability analysis which is especially important in healthcare. Second, EHR data is sensitive, and so submitting this to an API (e.g., as provided by OpenAI) is potentially problematic. In this work we show that despite the specialized domain, Flan-T5 (Chung et al., 2022; Wei et al., 2022) variants—which can fit on a single GPU and be run locally—can perform zero-shot feature extraction from EHR with reasonable accuracy.

Our contributions are as follows. (1) We propose and evaluate a method for extracting high-level interpretable features from clinical texts using Flan-T5 given corresponding prompts, and we evaluate the how well these extracted features align with ground truth. (2) We demonstrate that we can exploit LLM calibration to improve performance, allowing models that use inferred features to perform comparably to those using reference features. (3) We show that the resulting linear models' weights align with clinician-annotated notions of how features should impact a prediction. (4) We investigate the data- and feature- efficiency of our approach and find that it can achieve similar results with much less data and utilizes features efficiently.

Our promising initial results suggest several avenues for further exploration, e.g., modeling correlations between inferred features, probing the degree to which such predictors provide useful and reliable interpretability, and modifying trained linear models directly based on expert judgement.

## 2  Methods

We consider binary classification of patients on the basis of free text from their EHR data. A now standard approach to such tasks would entail adopting a neural language encoder $E$ to induce a fixed-length $d$-dimensional distributed representation of an input $x$—e.g., the [CLS] embedding in BERT (Devlin et al., 2018; Alsentzer et al., 2019) models—and feeding this forward to a linear classification head to yield a logit:

$$p(y = 1 \mid x) = \sigma(w^T E_{[\text{CLS}]}(x)) \qquad (1)$$

where $y \in \{0, 1\}$, $x \in \mathbb{R}^{L \times V}$, and $w \in \mathbb{R}^d$. (For BERT, $d = 768$.) A drawback of this approach is that it is not amenable to inspection. The prediction is made on the basis of a dense learned representation from a pre-trained network, and it is unclear which patient attributes give rise to the prediction. A simpler and (at least arguably) more interpretable

approach is to use a linear model defined over Bag-of-Words (BoW) representations:

$$p^{\text{BoW}}(y = 1|x) = \sigma(w^T x^{\text{BoW}}). \qquad (2)$$

This approach operates over (transformations of) token counts, and therefore the learned $w$ has a natural correspondence to words in the vocabulary. Linear models that operate over tens of thousands of word predictors occupy an intermediate space between the interpretability afforded by simpler, smaller models defined over high-level features and neural models which use opaque representations. For some clinical tasks, however, BoW with large vocabularies can be competitive with respect to downstream performance.

In this paper, we use instruction-tuned LLMs to perform zero-shot inference of intermediate, high-level, features $f \in \mathbb{R}^N$ using $N$ expert-specified prompt templates $t_1, ..., t_N$:

$$f_n^{\text{binary}} = \mathbb{I}[\text{argmax}_z(\text{LLM}(z|t_n(x))) = v_{\text{yes}}]$$
$$(3)$$

where $t_n(x)$ denotes the prompt obtained by populating the template $t_n$ with $x$, $z$ represents the next token after the prompt and $v_{\text{yes}}$ represents the index of the token "yes" in the vocabulary. We then use a simple linear model (with weights $w \in \mathbb{R}^N$) over these predicted features to predict the target:

$$p(y = 1) = \sigma(w^T f^{\text{binary}}). \qquad (4)$$

Predictions from the LLM for the high-level binary features will be imperfect, and are naturally associated with a confidence under the LLM: The probability of the token "yes" normalized by the mass assigned to *either* "yes" or "no". As a simple means of incorporating uncertainty in extracted features, we can use this continuous value in place of binary feature indicators. For feature $n$ (elicited using template $t_n$), the feature value $f_n^{\text{cont}}$ is:

$$\frac{\text{LLM}(z = v_{\text{yes}}|t_n(x))}{\text{LLM}(z = v_{\text{yes}}|t_n(x)) + \text{LLM}(z = v_{\text{no}}|t_n(x))}.$$
$$(5)$$

Inspired by previous work (Zhang et al., 2020), we split text into chunks of a particular maximum length and take the maximum (per-feature) of the feature values of the chunks as the final features.

## 3  Evaluation

To evaluate the proposed approach, we consider four tasks on publicly available MIMIC data to

permit reproducibility. We start with standard clinical predictive tasks in MIMIC-III (Johnson et al. 2016b,a; Goldberger et al. 2000, Section 3.1): Readmission, mortality, and phenotype prediction. For these we treat ICD codes as proxies for high-level features.[1] This allows us to evaluate the accuracy of the zero-shot extraction component, and to compare the performance of a linear model defined over the true ICD codes as compared to the same model when operating over inferred features.

We then consider X-ray report classification using the MIMIC-CXR dataset (Johnson et al. 2019b,a,c; Johnson et al.; Goldberger et al. 2000; Section 3.2). In this setting, we elicited queries for intermediate feature descriptions from a radiologist co-author. Queries take the form of questions that a radiologist might *a priori* believe to be relevant to the report classification categories defined in CheXpert (e.g., "Does this patient have a normal cardiac silhouette?"; see Appendix Table B for all examples). This demonstrates the flexibility of the approach—and the promise of allowing domain experts to specify features in natural language—but we are limited in our ability to evaluate the extraction performance directly in this case.

### 3.1 Clinical predictive tasks on MIMIC-III

For the three standard clinical prediction tasks we consider, we use ICD codes as proxies for high-level, interpretable features that one might want to infer. While this task is somewhat artificial, it allows us to evaluate how well the inferred features agree with the "true" features (i.e., ICD-code indicators). We use the top 10 most common ICD codes in the training set as features.[2] We ask the LLM: "Does this mean the patient has ?", where we replace  with the long description of the ICD code. To illustrate the flexibility of the approach, we also consider two custom features which one might expect to be informative: (1) Does the patient have a chronic illness? (2) Is the condition life-threatening?

**Readmission prediction** For 30-day readmission, we follow the task definition, setup, and data preprocessing outlined in (Huang et al., 2019).

**In-hospital Mortality prediction** This task involves predicting if a patient will pass during a hospital stay with notes from the first 48 hours. We adapt preprocessing code from (Zhang et al., 2020).

**Phenotype prediction** Zhang *et al.* (2020) also derive various phenotypes using ICD codes, and use these to define a phenotype prediction task which we also consider. (We again adapt their preprocessing code for this task.)

### 3.2 Chest X-ray report classification

We next consider chest X-ray report classification.[3] This allows us to draw upon the expertise of the radiologist co-author. We use the MIMIC-CXR dataset (Johnson et al., 2019b,a,c; Johnson et al.; Goldberger et al., 2000) with CheXpert labels (Irvin et al., 2019)—again to ensure reproducibility—and evaluate performance on a 12-label classification task.

Given that these labels can be automatically derived, the predictive task considered here is not in and of itself practically useful, but it is illustrative of how domain experts (here, a radiologist) can craft bespoke features for a given task. We use outputs of the CheXpert automated report labeler as "labels". We omit two downstream labels from our results—Consolidation and Pleural Effusion—because we included these as intermediate features instead under the advisement of our radiologist co-author, and it does not make sense to include labels for which there exists an exactly corresponding feature.[4] (The names of the 12 labels we predict are shown in Table 2.)

To infer intermediate features, we asked the radiologist co-author to provide natural binary questions they would ask to categorize radiology reports into the given classes (Appendix B). We refer to these questions as queries, and use them as prompts for the LLM to extract corresponding indicators (or probabilities) for each answer. Because these are novel predictors specified by a domain expert, we do not have "reference" features to compare to (as we did above where we treated ICD codes as high-level features). However, in Section 4.3 a domain expert assesses the degree to which learned coefficients for features align with clinical judgement.

---

[1] In practice one could of course use ICD codes directly as predictors; but this serves as an exemplary task using publicly available data to show the potential of the proposed approach in a way that also allows us to evaluate models that have access to "ground-truth" high-level features, i.e., the ICD codes.

[2] In preliminary experiments we found that including >10 ICD codes as predictors in linear models did not appreciably improve AUROC for the three tasks considered.

---

[3] We discard the images in this setting and only use text.

[4] We did use these extra two labels in training, but this only affects BERT, which was trained in a 'multi-task' way with a shared encoder used (and updated) across all labels. We think it unlikely that this would impact performance much.

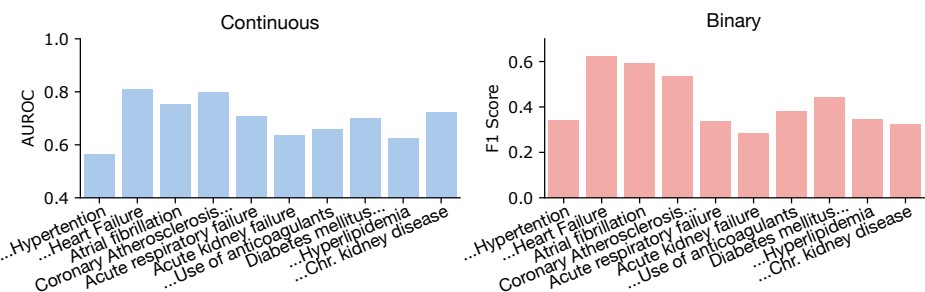

Figure 2: Feature extraction performance for the readmission prediction task. AUROC for continuous ICD code features (left) and F1 for binary ICD code features (right), compared to reference ICD codes.

## 3.3 Experimental details

To extract features, we use FLAN-T5-XXL (Roberts et al., 2022) with fp16 enabled (for speed) on a Quadro RTX 8000 with 46G of memory. We use a maximum chunk size of 512 (as described in section 2) and use a maximum of 4 chunks. To fit logistic regression models, we use the sklearn (Pedregosa et al., 2011) package's SGDClassifier with the logistic loss and the the default settings (this includes adding an intercept and an $\ell 2$ penalty). We show the full prompt template used for getting the features in appendix A.

## 4 Results

We aim to address the following research questions.

**Feature extraction (4.1)** How *accurate* are zero-shot extracted features, as compared to reference (manually extracted) predictors?

**Downstream classification performance (4.2)** How does classification based on exper our approach compare to black box and simple models on the ultimate downstream classification?

**Interpretability (4.3)** Do the inferred features permit intuitive interpretation, and do the resultant coefficients align with clinical expectations?

**Data and feature efficiency (4.4)** Do these features offer additional benefits in terms of data and/or feature efficiency?

## 4.1 Feature extraction

We first measure the accuracy of automatically inferred high-level features. In the case of ICD-code proxy features, we evaluate this directly using the actual ICD codes as reference labels with which to calculate precision, recall, and F1 for binary features, and AUROC for our continuous features (although we cannot do this for the two custom features considered). We present these metrics for

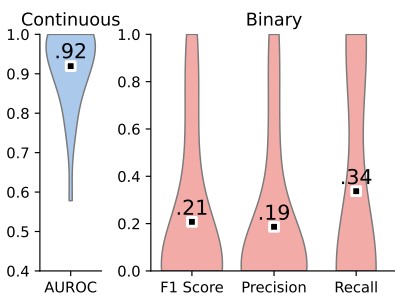

Figure 3: Feature inference performance on labeled Chest X-ray data. Histograms are over features. For continuous feature AUROCs, we omit features that do not have any positive labels in the 50 labeled examples. For F1 of binary features, we omit features that correspond to ill-defined precision (no positive predictions) and/or recall (no positive labels) scores are set to 0.

the readmission task in Figure 2.

For the radiology task, we have no reference features to use for evaluation. Therefore, our radiologist co-author annotated 50 test example reports with a set of features applicable to each report. This allows us to create binary labels for each feature and report, in turn allowing evaluation with the same metrics used above for binary and continuous feature encodings. There are 105 features—too many to present individually—so we report the feature scores in violin plots (Figure 3).

Zero-shot (feature) extraction performs reasonably well here in general, as suggested by prior work (Agrawal et al. 2022; although they used GPT-3, not Flan-T5). Scores are well above chance for MIMIC-III features. Given the zero-shot setting, they are not on par with supervised approaches for ICD code prediction (Mullenbach et al., 2018), but this may be partially due to a limitation of the ICD code reference features: ICD codes are known to be noisy (Kim et al., 2021; Boag et al., 2022), so even correctly inferred features may not align well

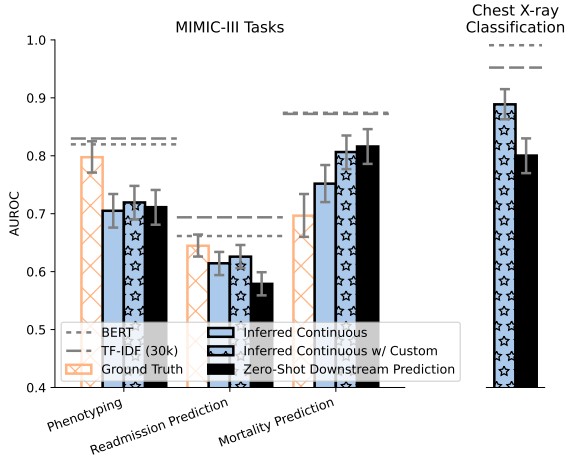

Figure 4: Downstream classification performance. We compare variants that use inferred continuous features with models that use "ground truth" features (here, ICD codes), and models that perform zero-shot prediction directly. We also show the performance of TF-IDF and BERT models (dashed horizontal lines) and 95% CIs.

with the labels. In contrast, our radiologist's direct annotation of the Chest X-ray report features reveal much higher AUROCs for continuous features. A novel aspect of this work, in contrast to (Agrawal et al., 2022), is that in the case of the chest X-ray task *the features are extracted using custom queries, specified as natural language questions provided by a domain expert (radiologist)*. While far from perfect, our results suggest that modern large instruction-tuned LMs are capable of inferring arbitrary clinically salient high-level features from EHR with some fidelity.

## 4.2 Downstream classification performance

Classification performance in and of itself is not our primary objective; rather, our aim is to design a method which allows for flexible construction of small and interpretable models using high-level features, with minimal expert annotation effort. But to contextualize the predictive performance of the models evaluated we also consider several baselines that offer varying degrees of "interpretability". Specifically, we evaluate fine-tuned variants of (a) Clinical BERT (Alsentzer et al., 2019), and (b) a logistic regression model defined over BoW (TF-IDF) features with varying vocabulary sizes. We report AUROCs for phenotyping and X-ray report classification, macro-averaged over labels.

To evaluate the degree to which zero-shot feature inference degrades performance, we also report results for a logistic regression model defined over *reference* ICD codes ("Ground Truth") for MIMIC tasks. And finally, we compare against direct zero-shot prediction of the *downstream classification* using FLAN-T5-XXL (Roberts et al., 2022).

Figure 4 demonstrates that across all tasks, the model variants with continuous intermediate features have significant signal (i.e., AUROC is significantly above chance 0.5).[5] Our method also performs comparably to or better than "Ground Truth" features for all tasks except Phenotyping, where the ICD code features (real more-so than inferred[6]) have an advantage because the phenotypes considered were derived from ICD codes.[7] We also see that the addition of just two custom queries does improve performance to varying degrees for MIMIC-III tasks relative to models that solely employ ICD-code queries, indicating that there is indeed a benefit that can be derived from employing natural language queries to predict intermediate features.

That said, making downstream predictions using these features performs worse than BERT and TF-IDF (30k) models. This is not surprising given that the number of features for our method is in the range of 10-105 as compared with 30k for TF-IDF and 100k+[8] for ClinicalBERT.

Zero-shot prediction of the downstream task performs worse than (supervised) linear models on top of inferred features on Readmission prediction and Chest X-ray report classification, equivalently on Phenotyping, and slightly better on Mortality prediction. However, such predictions are completely blackbox (see Section 5 for discussion around interpretability of zero-shot extracted features.) Finally, we also find that using binary features instead of continuous features degrades performance significantly (Table 1); calibrating (un)certainty helps.

When manually inspecting some example radiology reports, it became apparent that downstream labels are often verbalized in the radiology report.

| w/ Custom | Pheno. | Readmis. | Mort. | CXR Class. |
|---|---|---|---|---|
| ✗ | -.054 | -.025 | -.052 | - |
| ✔ | -.050 | -.023 | -.088 | -.060 |

Table 1: Difference in AUROC between using Binary and Continuous features.

---

[5]Expanded set of results in Apppendix Table C.1.

[6]Inferred codes are not expected to fully align with noisy ICD codes (see section 4.1).

[7]Performance is not 100% because only a subset of the ICD codes used for phenotyping were used as features.

[8]approximately, after the embedding layer

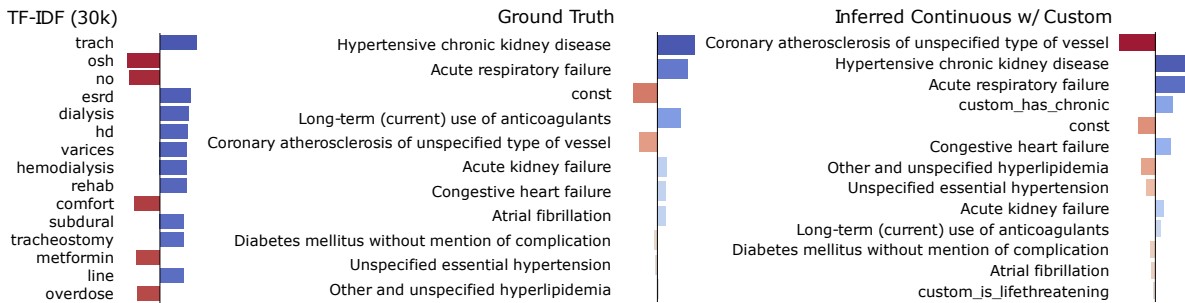

Figure 5: Linear model coefficients for readmission prediction. Blue and red indicate features that support or refute the label, respectively, and "const" refers to the values of the intercept. Though many feature weights (e.g. "Hypertensive Chronic Kidney Disease") make a lot of sense in the context of readmission prediction, "Coronary atherosclerosis [...]," which we see in the inferred features, does not make sense as a top feature. We suspect this is due to imperfect feature inference as it is not a top feature when using the ground truth features.

|  | P@1 | P@5 | P@10 | P@20 | AUC |
|---|---|---|---|---|---|
| No Finding | 1.00 | 0.40 | 0.20 | 0.10 | 0.62 |
| Enlar. Cardiom. | 1.00 | 0.20 | 0.10 | 0.05 | 1.00 |
| Cardiomegaly | 1.00 | 0.80 | 0.60 | 0.35 | 0.61 |
| Edema | 0.00 | 0.20 | 0.30 | 0.20 | 0.62 |
| Pneumonia | 0.00 | 0.20 | 0.20 | 0.10 | 0.64 |
| Atelectasis | 0.00 | 0.20 | 0.20 | 0.10 | 0.69 |
| Pneumothorax | 1.00 | 0.60 | 0.30 | 0.15 | 0.84 |
| Fracture | 1.00 | 0.40 | 0.30 | 0.15 | 0.74 |
| Lung Lesion | 0.00 | 0.60 | 0.40 | 0.25 | 0.80 |
| Lung Opacity | 0.00 | 0.00 | 0.00 | 0.00 | 0.29 |
| Pleural Other | 0.00 | 0.20 | 0.40 | 0.20 | 0.73 |
| Support Devices | 0.00 | 0.40 | 0.30 | 0.30 | 0.49 |
| Average | 0.42 | 0.35 | 0.27 | 0.16 | 0.67 |

Table 2: Precisions and AUCs of **learned feature rankings** on the Chest X-ray classification task, evaluated against *a priori* relevancy judgements per class provided by our radiologist collaborator. The model was *not* trained to rank features but nevertheless implicitly learned feature importance that aligns with intuition.

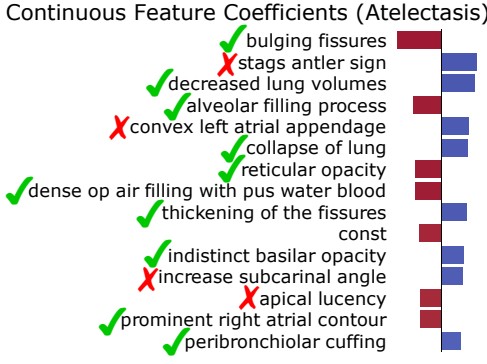

Figure 6: Linear model coefficients for predicting **Atelectasis** in the Chest X-ray Dataset using the top 15 continuous features. "const" represents a constant bias feature and is therefore not annotated. ✔ and ✗ denote post-hoc annotations of the feature coefficient as aligning or not aligning with clinical expectations, respectively. Most features (both with negative and positive coefficients) do align.

This makes sense given that the CheXpert labeler needs to extract these from the report, and intermediate features are therefore not explicitly modeled. This gives TF-IDF and BERT models a particular advantage over inferred feature models that is unlikely to exist for natural tasks (which are not defined by an automated labeler, and where intermediate features will probably be important).

### 4.3 Interpretability

Given our primary motivation to offer interpretability—specifically via small linear models over a small number of high-level features—we next investigate how well learned coefficients align with domain expert judgement.

The radiologist who specified features for the chest X-ray dataset also indicated for which task(s)

they judged each feature to be predictive. Consequently, we have a label for each of the 105 features that indicates whether the domain expert believed the feature to be likely supportive of, or likely not relevant to, each of the 12 classes defined in the chest X-ray task. We use these "feature labels" to measure the degree to which the learned weights agree with the domain knowledge that they are intended to encode. In particular, we rank each feature by coefficient and compute precision at $k$ and AUC for all 12 classes (Table 2). For all labels except Lung Opacity and Support Devices, the rank of features in terms of relevance consistently agrees with expert judgement (AUC > .5).

As further evidence of the greater interpretability afforded by the proposed approach, we report the top features for readmission prediction in Figure 5.

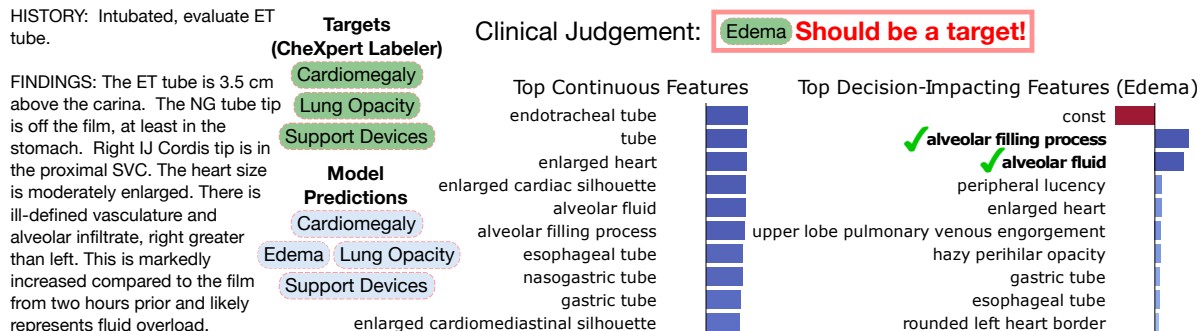

Figure 7: Qualitative example of features and feature influence for predicting Edema in the case of an ostensible "false positive". We selected this example because on cursory inspection Edema would seem to be applicable here. We presented this to our radiologist co-author who confirmed that *this report should in fact be labeled positive for Edema*. We show the report and the downstream reference and predicted labels on the left. In the middle, we report top raw feature values, and on the right we show the top scores, i.e., feature values times the corresponding coefficients for Edema. **Bolded** features were confirmed by the radiologist as aligning with clinical judgement.

Here we compare the top-ranked features in linear models using TF-IDF (left), reference ICD codes (the "ground truth" high-level features here; center), and inferred high-level features (right). The top positive inferred high-level features align with the top positive reference ICD code features.

For high-level features, coefficient magnitude mass is concentrated on the very top features, whereas in the TF-IDF case mass is more uniformly distributed. This held across many of the tasks and labels (see Appendix Table C.2), and is likely an artifact of having many more TF-IDF features. It renders such models difficult to interpret. We also see that the custom "has_chronic" feature is among the top features and the "is_lifethreatening" feature is at the bottom, aligning with intuition.

We also consider the coefficients for chest X-ray report classification, enlisting our radiologist co-author to annotate these. It is important to do this analysis post-hoc because the initial feature annotations used for the metrics in Table 2 are not necessarily exhaustive. Figure 6 reports assessments for the linear model coefficients for "atelectasis". Most of the feature influences agree with expectations.

Interestingly, this analysis indicates the presence of certain known reporting biases present in the reports. For example, the feature "apical lucency" specifically indicates possible pneumothorax, a cause of passive atelectasis, and so rationally should support the 'atelectasis' label, but is weighted to *refute* the label. We speculate that this reflects 'satisfaction of search' bias, and other closely aligned reporting biases; pneumothorax is such a critically important condition that radiologists reporting pneumothorax will in many cases

not spend time searching for, or reporting the associated atelectasis, which in this case is a secondary feature of far lower independent importance.

Figure 6 shows an example illustrating how a clinician might inspect a classification. Because the predicted "edema" label disagreed with the CheXpert labeler, our radiologist collaborator reviewed this case to determine which features led to this ostensible "false positive". While inspecting the features for a source of error, they determined that the top positive features ("alveolar filling process", "alveolar fluid") accurately described the report and could support Edema, but more commonly indicate pneumonia or other causes. Subsequently, while reviewing the report, the radiologist concluded that **the model prediction of edema was actually correct**; the CheXpert label was a false negative. In this case, the inferred features did correctly influence the 'edema/no edema' classification.

Appendix Figure C.2 shows an example where a feature influenced the model *incorrectly*, even though the model made the correct prediction. These examples illustrate two of many ways in which our approach might facilitate model interpretation and debugging.

### 4.4 Data- and feature- efficiency

Finally, we consider how our model fares compared to baselines in terms of data efficiency. Specifically, we construct learning curves by increasing the percent of the available data used to train models. In Figure 8, we see that the performance of small models plateaus with relatively minimal supervision. At low levels of supervision such models are competitive with—and in some cases better

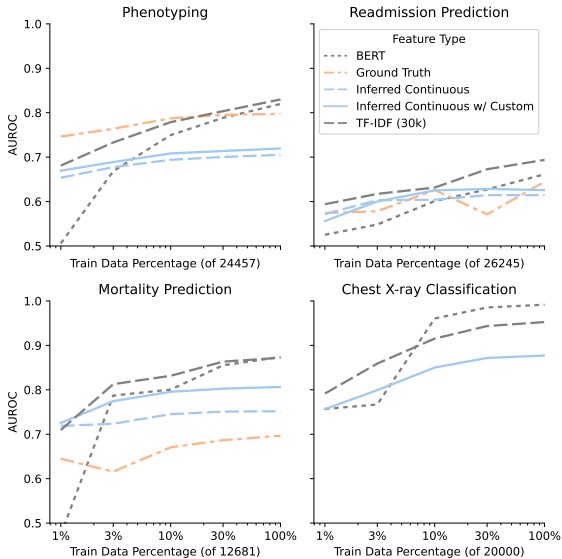

Figure 8: Learning curves for different methods.

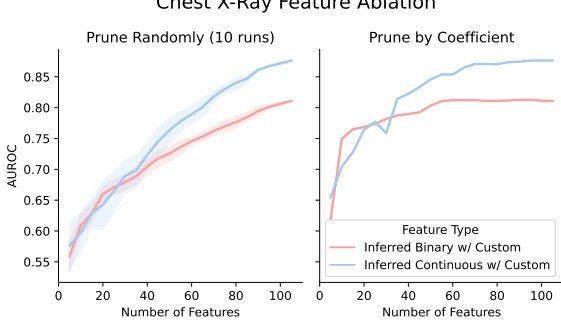

Figure 9: Feature ablation for Chest X-ray classification. After training, we explore how pruning features randomly (left) or based on coefficient magnitudes (averaged over all classes) affects performance.

than—larger models (e.g., based on Clinical BERT representations), which benefit more from additional supervision. We again emphasize, however, that using more complex representations comes at the expense of interpretability; this is true even for TF-IDF (see Figure 5).

We also explore the distribution of information among the features by pruning features from our continuous model after training in Figure 9. If we prune features randomly, we see a curve that indicates that we have not saturated our performance, and adding additional features will likely increase performance further. In fact, while annotating 50 examples on the test set, our radiologist co-author noted many features that were not included that would likely increase performance.

If we prune features with the smallest-magnitude coefficients first, we see that—in contrast to the model using binary features—the continuous feature model has a much more rounded curve, indicating that there are a large number of features that are actively contributing to a performance increase. We also note that we use dramatically fewer inferred features compared to TF-IDF. Indeed, as we report in Appendix Table C.1, if we limit TF-IDF to a vocabulary size of 100, using continuous inferred features outperforms TF-IDF on all tasks.

## 5  Discussion

Small linear models over well-defined features offer transparency in that one can read off coefficients for predictors and inspect which most influ-

enced the output for a given instance (e.g., Figure 7). This is in stark contrast to deep neural network models (like Clinical BERT; Alsentzer et al. 2019), which operate over dense learned embeddings. However, in healthcare high-level features must often be manually extracted from unstructured EHR. In this work we proposed to allow domain experts to define high-level predictors with natural language queries that are used to infer feature values via LLMs. We then define small linear models over the resulting features.

The promise of this approach is that it enables one to capitalize on LLMs while still building models that provide interpretability. Our approach allows domain experts to craft features that align with questions a clinician might seek to answer in order to make a diagnosis and then use the coefficients from a trained linear model to determine which features inform predictions. Using such "abstract" features in place of word frequency predictors confers greater interpretability (Figure 5).

## 6  Conclusions

We have proposed using large language models (LLMs) to infer high-level features from unstructured Electronic Health Records (EHRs) to be used as predictors in small linear models. These high-level features can be specified in arbitrary natural language queries composed by domain experts, and extracted using LLMs without explicit supervision.

On three clinical predictive tasks, we showed that this approach yields performance comparable to that achieved using "reference" high-level features (here, ICD codes). On a fourth task (X-ray report classification), we enlisted a radiologist collaborator to provide high-level features in natural lan-

guage. We showed a model using these features as automatically extracted via an LLM realized strong predictive performance, and—more importantly—provided varieties of model transparency which we confirmed aligned with clinical judgement, and which provided insights to the domain expert.

This work demonstrates the promise of using LLMs as high-level feature extractors for small linear models, which may admit varieties of interpretability. However, this is complicated by the issues discussed in the Limitations section, below. We believe that more research into this hybrid approach is therefore warranted.

## Limitations

While the initial results reported above are encouraging, they also raise several questions as to how one should design queries to yield intermediate features that are interpretable, informative, and verifiable. One limitation of using LLMs to generate high-level features is that inference for these features remains opaque. This inherent limitation suggests a few immediate research questions for future work.

**The slippery "interpretability" of inferred features; Is there a trade-off between interpretability and predictive power?**   Not every query informative of the downstream prediction task will necessarily aid interpretability. For example, when predicting mortality, the response to the query *"Is the patient at risk of death?"* will likely correlate strongly with the downstream task (as can be seen in Figure 4, which reports performance for zero-shot prediction). But this query essentially paraphrases the original downstream task, and so does little to aid interpretability. It instead simply shifts the problem to explaining what elements of the EHR give rise to the predicted "feature". This suggests that expert queries should be written to elicit features that correlate with, but are distinct from, the prediction task.

**How do we know whether predicted features are faithful to their queries?**   A related complication to this approach is that zero-shot feature extraction will not be perfect. While results for ICD code proxies indicate good correlation, we will in general not be able to easily verify whether the indicator that is extracted by an LLM is in fact predicting the feature intended. The interpretability of the resultant model will therefore depend on

the degree to which the LLM inferences align with the domain experts' intent regarding the respective features.

This suggests that one might want to design queries for high-level features that, whenever possible, are easily verifiable upon inspection of the EHR. Asking if a patient is at risk of death is not directly verifiable, because it is necessarily a prediction; asking if the patient has an "enlarged heart" (for example) probably is. However, even using verifiable features does **not** guarantee that clinicians will hastily confirm or remain unbiased by unverified features in practice.

**Can we manually intervene when a model learns a counter-intuitive dependency?**   Because we exploit features that have a direct relationship to domain expert knowledge, it might be possible to draw upon their judgement to intervene when a model learns a dependence on a predictor in way that does not align with expectations (e.g. Figures 6 and C.2). This may happen because either: (1) the feature itself was incorrectly inferred, (2) there exists some spurious correlation in the data that does not hold in general. A less likely possibility is that (3) the expert prior was simply incorrect. Future work might study if pruning this feature to "correct" the model improves generalization.

## Ethics Statement

Our approach raises ethical concerns because the extracted features may be incorrect. Though this work does show promise in terms of the interpretations aligning with clinical expectations, it is far from ready to deploy because there is a danger that clinicians will make incorrect assumptions even if warned of the model's potential for making mistakes both in producing features. Just because the explanation of the model makes sense clinically does not mean the underlying features for an instance are factual. Models like this may also play into clinician bias given that the model is trained data produced by clinicians. Therefore, we present this work only for scientific exploration, and it is not intended to be used in a deployed system in its current form.

## Acknowledgements

We acknowledge partial funding for this work by National Library of Medicine of the National Institutes of Health (NIH) under award numbers

R01LM013772 and R01LM013891. The work was also supported in part by the National Science Foundation (NSF) grant 1901117. The content is solely the responsibility of the authors and does not necessarily represent the official views of the NIH or the NSF.

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

## A  Full Prompt

Table A.1: The full prompts incorporate the input text and the questions, described in section 3 and appendix Table B according to the following templates.

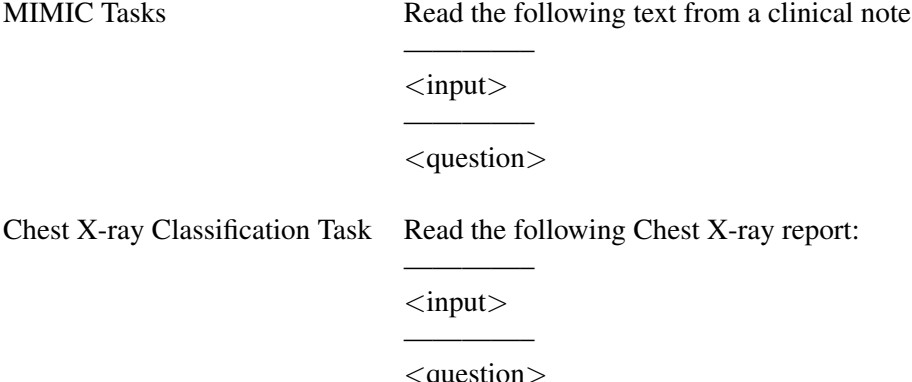

|  |  |
| --- | --- |
| MIMIC Tasks | Read the following text from a clinical note:
————
<input>
————
<question> |
| Chest X-ray Classification Task | Read the following Chest X-ray report:
————
<input>
————
<question> |

## B  Hand-crafted Feature Queries for Chest X-ray Dataset

Table B.2: Features grouped by the labels they may support. (Features may be written under more than one label.)

| **No Finding** | |
| --- | --- |
| normal | Is this patient normal? |
| clear lungs | Does this patient have clear lungs? |
| normal cardiac silhouette | Does this patient have a normal cardiac silhouette? |
| sharp costophrenic angles | Does this patient have sharp costophrenic angles? |
| **Enlarged Cardiomediastinum** | |
| enlarged cardiomediastinal silhouette | Does this patient have an enlarged cardiomediastinal silhouette? |
| **Cardiomegaly** | |
| enlarged cardiomediastinal silhouette | Does this patient have an enlarged cardiomediastinal silhouette? |
| increased cardiothoracic ratio | Does this patient have an increased cardiothoracic ratio? |
| prominent right atrial contour | Does this patient have a prominent right atrial contour? |
| enlarged heart | Does this patient have an enlarged heart? |
| globular cardiac silhouette | Does this patient have a globular cardiac silhouette? |
| rounded left heart border | Does this patient have a rounded left heart border? |
| uplifted cardiac apex | Does this patient have an uplifted cardiac apex? |
| double density | Does this patient have a double density? |
| splaying of carina | Is there splaying of the carina? |
| increase subcarinal angle | Does this patient have an increased subcarinal angle? |
| convex left atrial appendage | Does this patient have a convex left atrial appendage? |
| third mogul sign | Does this patient have a third mogul sign? |
| superior displacement of left mainstem bronchus | Does this patient have superior displacement of left mainstem bronchus? |

| | |
|---|---|
| rounded cardiac apex | Does this patient have a rounded cardiac apex? |
| shmoo sign | Does this patient have a shmoo sign? |
| hoffman rigler sign | Does this patient have a Hoffman-Rigler sign? |

**Edema**

| | |
|---|---|
| upper lobe pulmonary venous engorgement | Does this patient have upper lobe pulmonary venous engorgement? |
| stags antler sign | Does this patient have a stag's antler sign? |
| bilateral opacity | Does this patient have bilateral opacity? |
| symmetric perihilar opacity | Does this patient have a symmetric perihilar opacity? |
| bat wing opacity | Does this patient have a bat wing opacity? |
| enlarged cardiac silhouette | Does this patient have an enlarged cardiac silhouette? |
| increased cardiothoracic ratio | Does this patient have an increased cardiothoracic ratio? |
| peribronchiolar cuffing | Does this patient have peribronchiolar cuffing? |
| hazy perihilar opacity | Does this patient have a hazy perihilar opacity? |
| septal thickening | Does this patient have septal thickening? |
| kerley b lines | Does this patient have Kerley B lines? |
| thickening of the fissures | Is there a thickening of the fissures? |
| fluid in the fissures | Does this patient have fluid in the fissures? |
| pleural effusion | Does this patient have a pleural effusion? |

**Pneumonia**

| | |
|---|---|
| reticular opacity | Does this patient have a reticular opacity? |
| hazy opacity | Does this patient have a hazy opacity? |
| consolidation | Does this patient have consolidation? |
| segmental opacity | Does this patient have a segmental opacity? |
| lobar opacity | Does this patient have a lobar opacity? |
| bulging fissures | Does this patient have bulging fissures? |
| cavitation | Does this patient have cavitation? |

**Atelectasis**

| | |
|---|---|
| subsegmental linear basilar opacity | Does this patient have a subsegmental linear basilar opacity? |
| linear basilar opacity | Does this patient have a linear basilar opacity? |
| subsegmental crescentic basilar opacity | Does this patient have a subsegmental crescentic basilar opacity? |
| crescentic basilar opacity | Does this patient have a crescentic basilar opacity? |
| decreased lung volumes | Does this patient have decreased lung volumes? |
| indistinct basilar opacity | Does this patient have indistinct basilar opacities? |

**Pneumothorax**

| | |
|---|---|
| apical lucency | Does this patient have an apical lucency? |
| peripheral lucency | Does this patient have a peripheral lucency? |
| collapse of lung | Is there a collapse of a lung? |
| visible visceral pleura | Does this patient have visible visceral pleura? |

**Fracture**

| | |
|---|---|
| lucency in a rib or clavicle | Does this patient have a lucency in a rib or clavicle? |
| deformity of a rib or clavicle | Does this patient have a deformity of a rib or clavicle? |
| wedge deformity of a vertebra | Does this patient have a wedge deformity of a vertebra? |

| | |
|---|---|
| step off of a rib or clavicle | Does this patient have a step-off of a rib or clavicle? |

### Lung Lesion

| | |
|---|---|
| rounded opacity | Does this patient have a rounded opacity? |
| pulmonary calcification | Does this patient have an pulmonary calcification? |
| cavitation | Does this patient have cavitation? |
| abnormal high density | Does this patient have an abnormal high density? |
| opacity with indistinct borders | Does this patient have an opacity with indistinct borders? |
| spiculated opacity | Does this patient have a spiculated opacity? |
| hazy opacity | Does this patient have a hazy opacity? |

### Lung Opacity

| | |
|---|---|
| alveolar blood | Is there alveolar blood? |
| alveolar fluid | Is there alveolar fluid? |

### Pleural Other

| | |
|---|---|
| pleural empyema | Is there pleural empyema? |
| hemothorax | Does the patient have hemothorax? |
| pleural tumor | Is there pleural tumor? |
| pleural thickening | Is there pleural tumor? |
| calcificied pleural plaques | Does this patient have a calcified pleural plaque? |
| foreign body in pleural space | Is there a foreign body in the pleural space? |

### Support Devices

| | |
|---|---|
| surgical clip | Does this patient have a surgical clip? |
| metallic density | Does this patient have a metallic density? |
| curvilinear density | Does this patient have a curvilinear density? |
| foreign body | Does this patient have a foreign body? |
| metal lead | Does this patient have a metal lead? |
| wire | Does this patient have a wire? |
| tube | Does this patient have a tube? |
| stent | Does this patient have a stent? |
| endotracheal tube | Does this patient have an endotracheal tube? |
| chest tube | Does this patient have a chest tube? |
| central venous catheters | Does this patient have a central venous catheter? |
| picc line | Does this patient have a PICC line? |
| tracheal stent | Does this patient have a tracheal stent? |
| coronary stent | Does this patient have a coronary stent? |
| aortic stent | Does this patient have an aortic stent? |
| arterial stent | Does this patient have an arterial stent? |
| pacemaker | Does this patient have a pacemaker? |
| icd | Does this patient have an ICD? |
| aortic balloon pump | Does this patient have an aortic balloon pump? |
| lvad | Does this patient have an lvad? |
| clamshell closure device | Does this patient have a clamshell closure device? |
| SVC stent | Does this patient have an SVC stent? |
| IVC stent | Does this patient have an IVC stent? |
| IVC filter | Does this patient have an IVC filter? |

### Other Features

| | |
|---|---|
| alveolar hemorrhage | Is there alveolar hemorrhage? |
| dense op air filling with pus water blood | Does this patient have a dense opacity that suggests the airspace filling with pus, water, or blood? |
| esophageal tube | Does this patient have an esophageal tube? |

| | |
|---|---|
| g-tube | Does this patient have a g-tube? |
| gastric tube | Does this patient have a gastric tube? |
| loculated pleural effusion | Is there a loculated pleural effusion? |
| nasogastric tube | Does this patient have a nasogastric tube? |

|  | Phenotyping | Readmission | Mortality | Chest X-ray |
|---|---|---|---|---|
| BERT | $0.820 \pm 0.024$ | $0.661 \pm 0.019$ | $0.874 \pm 0.025$ | $0.991 \pm 0.008$ |
| TF-IDF (30) | $0.624 \pm 0.031$ | $0.573 \pm 0.020$ | $0.692 \pm 0.034$ | $0.760 \pm 0.033$ |
| TF-IDF (100) | $0.697 \pm 0.030$ | $0.613 \pm 0.020$ | $0.787 \pm 0.031$ | $0.881 \pm 0.027$ |
| TF-IDF (1k) | $0.813 \pm 0.025$ | $0.668 \pm 0.019$ | $0.863 \pm 0.025$ | $0.952 \pm 0.014$ |
| TF-IDF (30k) | $0.830 \pm 0.024$ | $0.694 \pm 0.019$ | $0.872 \pm 0.024$ | $0.952 \pm 0.013$ |
| Ground Truth | $0.798 \pm 0.027$ | $0.645 \pm 0.019$ | $0.697 \pm 0.037$ | - |
| Inferred Binary | $0.651 \pm 0.031$ | $0.589 \pm 0.020$ | $0.699 \pm 0.035$ | - |
| Inferred Binary w/ Custom | $0.669 \pm 0.031$ | $0.603 \pm 0.020$ | $0.719 \pm 0.033$ | $0.828 \pm 0.034$ |
| Inferred Continuous | $0.705 \pm 0.029$ | $0.614 \pm 0.020$ | $0.752 \pm 0.032$ | - |
| Inferred Continuous w/ Custom | $0.719 \pm 0.029$ | $0.626 \pm 0.020$ | $0.806 \pm 0.029$ | $0.889 \pm 0.026$ |
| Zero-Shot Downstream | $0.711 \pm 0.030$ | $0.579 \pm 0.020$ | $0.816 \pm 0.030$ | $0.800 \pm 0.030$ |

Table C.1: Expanded results of Downstream Classification Performance with 95% Confidence Intervals.

|  | Phenotype | Readmission | Mortality | Chest X-ray |
|---|---|---|---|---|
| TF-IDF (30) | 3.05 | 3.41 | 1.00 | 1.98 |
| TF-IDF (100) | 3.84 | 4.54 | 2.67 | 1.79 |
| TF-IDF (1k) | 4.63 | 6.84 | 6.24 | 1.86 |
| TF-IDF (30k) | 9.50 | 10.30 | 10.27 | 4.17 |
| Ground Truth | 1.30 | 2.39 | 2.00 | - |
| Inferred Binary | 1.96 | 2.39 | 1.85 | - |
| Inferred Binary w/ Custom | 2.03 | 2.56 | 1.50 | 3.98 |
| Inferred Continuous | 1.65 | 2.31 | 1.97 | - |
| Inferred Continuous w/ Custom | 1.75 | 2.51 | 1.13 | 3.08 |

Table C.2: **How is the magnitude mass distributed across coefficients?** We report the **entropy** of the distribution described by the coefficient magnitudes: $H(\text{softmax}(|\mathbf{c}|))$ where $\mathbf{c}$ represents the vector of coefficients for a model and $|\cdot|$ takes the absolute value of each element. (Entropy is averaged over labels in the case of Phenotype and Chest X-ray tasks.) This measure gives some insight into how uniform the coefficient magnitudes are. TF-IDF (30k) has much higher entropy across the board indicating that many more features have high magnitude in this model.

## C   Extra Figures and Tables

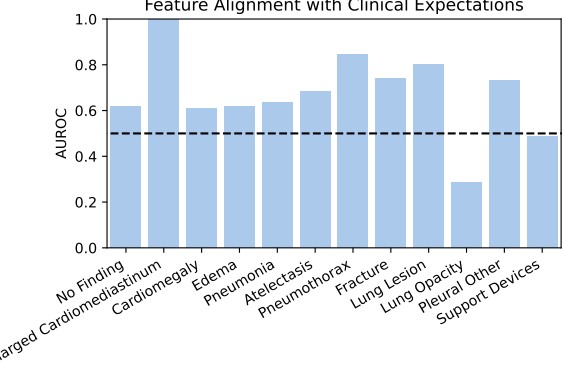

Figure C.1: Using a set of features annotated as supporting a particular label, we plot the AUROC obtained by using the coefficient values (from the continuous features model) to predict this feature list for each label.

Table C.3: **AUC per Phenotyping label.**

| | BERT | TF-IDF (30k) | Ground Truth | Inferred Continuous | Inferred Continuous w/ Custom | Zero-Shot Downstream |
|---|---|---|---|---|---|---|
| Conduction Disorders | 0.856 | 0.847 | 0.712 | 0.678 | 0.676 | 0.758 |
| Pneumonia (Except That Caused By Tuberculosis Or Sexually Transmitted Disease) | 0.809 | 0.834 | 0.744 | 0.709 | 0.713 | 0.733 |
| Disorders Of Lipid Metabolism | 0.785 | 0.763 | 1.000 | 0.705 | 0.723 | 0.639 |
| Acute Myocardial Infarction | 0.867 | 0.862 | 0.785 | 0.788 | 0.796 | 0.805 |
| Other Lower Respiratory Disease | 0.696 | 0.719 | 0.684 | 0.580 | 0.630 | 0.649 |
| Pleurisy; Pneumothorax; Pulmonary Collapse | 0.663 | 0.722 | 0.583 | 0.593 | 0.627 | 0.661 |
| Any-Chronic | 0.874 | 0.867 | 0.951 | 0.779 | 0.784 | 0.659 |
| Diabetes Mellitus With Complications | 0.863 | 0.867 | 0.652 | 0.774 | 0.776 | 0.831 |
| Cardiac Dysrhythmias | 0.846 | 0.838 | 0.905 | 0.797 | 0.799 | 0.793 |
| Any-Acute | 0.819 | 0.834 | 0.819 | 0.715 | 0.726 | 0.515 |
| Coronary Atherosclerosis And Other Heart Disease | 0.870 | 0.868 | 0.964 | 0.820 | 0.838 | 0.801 |
| Hypertension With Complications And Secondary Hypertension | 0.873 | 0.870 | 0.864 | 0.749 | 0.769 | 0.670 |
| Septicemia (Except In Labor) | 0.874 | 0.883 | 0.748 | 0.732 | 0.728 | 0.782 |
| Respiratory Failure; Insufficiency; Arrest (Adult) | 0.882 | 0.884 | 0.985 | 0.787 | 0.784 | 0.780 |
| Any-Disease | 0.903 | 0.897 | 0.934 | 0.812 | 0.826 | 0.632 |
| Congestive Heart Failure; Non-hypertensive | 0.837 | 0.830 | 0.979 | 0.767 | 0.770 | 0.704 |
| Chronic Obstructive Pulmonary Disease And Bronchiectasis | 0.777 | 0.789 | 0.670 | 0.658 | 0.670 | 0.716 |
| Complications Of Surgical Procedures Or Medical Care | 0.712 | 0.756 | 0.589 | 0.569 | 0.581 | 0.701 |
| Acute And Unspecified Renal Failure | 0.830 | 0.848 | 0.956 | 0.703 | 0.714 | 0.646 |
| Shock | 0.873 | 0.872 | 0.756 | 0.715 | 0.726 | 0.694 |
| Other Upper Respiratory Disease | 0.831 | 0.833 | 0.642 | 0.686 | 0.698 | 0.737 |
| Fluid And Electrolyte Disorders | 0.734 | 0.760 | 0.691 | 0.648 | 0.650 | 0.618 |
| Other Liver Diseases | 0.828 | 0.855 | 0.661 | 0.641 | 0.669 | 0.730 |
| Gastrointestinal Hemorrhage | 0.855 | 0.887 | 0.583 | 0.628 | 0.636 | 0.779 |
| Diabetes Mellitus Without Complication | 0.700 | 0.723 | 0.971 | 0.719 | 0.720 | 0.638 |
| Acute Cerebrovascular Disease | 0.907 | 0.936 | 0.677 | 0.628 | 0.709 | 0.841 |
| Essential Hypertension | 0.729 | 0.720 | 0.996 | 0.627 | 0.634 | 0.588 |
| Chronic Kidney Disease | 0.857 | 0.869 | 0.830 | 0.736 | 0.771 | 0.817 |

Table C.4: **AUC per CheXpert label.**

| | BERT | TF-IDF (30k) | Inferred Continuous w/ Custom | Zero-Shot Downstream |
|---|---|---|---|---|
| Atelectasis | 0.997 | 0.968 | 0.888 | 0.944 |
| Cardiomegaly | 0.997 | 0.957 | 0.913 | 0.876 |
| Edema | 0.997 | 0.961 | 0.910 | 0.954 |
| Enlarged Cardiom. | 0.988 | 0.904 | 0.764 | 0.592 |
| Support Devices | 0.996 | 0.965 | 0.925 | 0.826 |
| Pneumothorax | 0.990 | 0.950 | 0.911 | 0.868 |
| Fracture | 0.999 | 0.987 | 0.948 | 0.977 |
| Pneumonia | 0.982 | 0.921 | 0.795 | 0.799 |
| Pleural Other | 0.995 | 0.965 | 0.804 | 0.566 |
| No Finding | 0.987 | 0.944 | 0.906 | 0.230 |
| Lung Opacity | 0.994 | 0.944 | 0.899 | 0.858 |
| Lung Lesion | 0.976 | 0.965 | 0.862 | 0.804 |

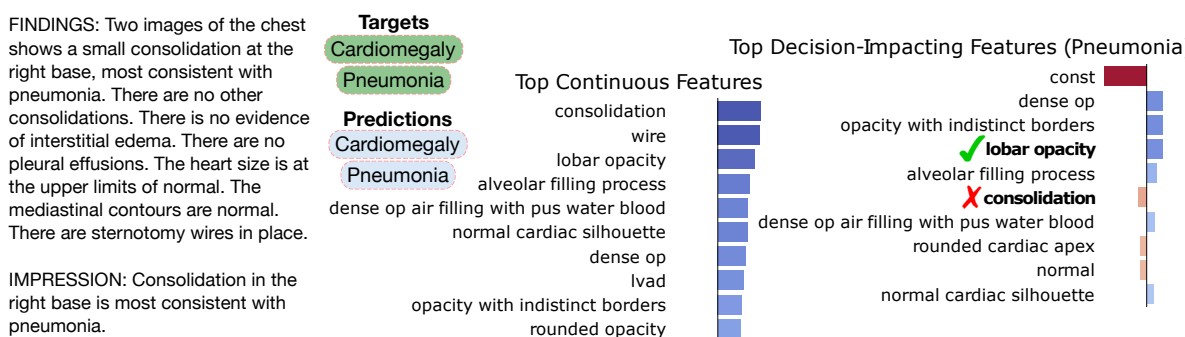

FINDINGS: Two images of the chest shows a small consolidation at the right base, most consistent with pneumonia. There are no other consolidations. There is no evidence of interstitial edema. There are no pleural effusions. The heart size is at the upper limits of normal. The mediastinal contours are normal. There are sternotomy wires in place.

IMPRESSION: Consolidation in the right base is most consistent with pneumonia.

**Targets**
Cardiomegaly
Pneumonia

**Predictions**
Cardiomegaly
Pneumonia

Top Continuous Features

consolidation
wire
lobar opacity
alveolar filling process
dense op air filling with pus water blood
normal cardiac silhouette
dense op
lvad
opacity with indistinct borders
rounded opacity

Top Decision-Impacting Features (Pneumonia)

const
dense op
opacity with indistinct borders
✔ **lobar opacity**
alveolar filling process
✗ **consolidation**
dense op air filling with pus water blood
rounded cardiac apex
normal
normal cardiac silhouette

Figure C.2: Qualitative Example of Features and Feature Influence for Pneumonia. (Similar to Figure 7.) This seems to correctly predict Pneumonia, but this happens in spite of consolidation being incorrectly identified by the linear model as not supporting Pneumonia. These plots make clear that this is not a mistake of the feature extractor at inference time because consolidation is among the top Features. The mistake comes from the linear model having a negative coefficient for consolidation.