# OpenReview forum: "CHiLL: Zero-shot Custom Interpretable Feature Extraction from Clinical Notes with Large Language Models"
_EMNLP/2023/Conference — EMNLP 2023 Findings_

### Official Review · Reviewer_J7J2 · 2023-07-31

**Soundness:** 3

**Excitement:**

3: Ambivalent: It has merits (e.g., it reports state-of-the-art results, the idea is nice), but there are key weaknesses (e.g., it describes incremental work), and it can significantly benefit from another round of revision. However, I won't object to accepting it if my co-reviewers champion it.

**Paper Topic And Main Contributions:**

I'm not sure whether my understanding is correct because the methodology section of this study is a little bit hard to follow. It seems this study proposed a two-phase model. In the first phase, the author adopted a large language model (LLM, Flan-T5-XXL) to extract a series of preset features in a zero-shot manner. Then,  they trained a simple linear model to generate labels via the extracted features in a supervised manner. As the simple linear model is intrinsically interpretable, the classification result can be regarded as interpretable.

**Reasons To Accept:**

From the point of my view, the main contribution of this study is that the author and their clinical collaborator derived a detailed, hand-crafted prompt set to extract features from Chest X-ray Dataset (Supplementary A, B). I believe these prompts can significantly help the community to establish a structured dataset from unstructured medical text.

**Reasons To Reject:**

The main drawback of this study is that the authors did not prove their model is better than the TF-IDF-based BoW model or other traditional models.

Of note, Medical text classification basically does not require a model to understand the language. The task performance basically relies on the existence of several key-word. Therefore, we can observe that the BERT-based model did not obtain significantly better performance than TF-IDF (BoW) model, and the performance of proposed model even obtained worse performance than the TF-IDF model (Figure 4, 8). Although the author claimed performance is not their primary objective, it is disappointing that such a computationally expensive model only obtained a worse performance than a model proposed several decades ago. Meanwhile, there is also a topic model (i.e., LDA) or neural network-based topic model that can extract interpretable features and be applied to downstream tasks. It will be better if the author can include them as baselines.

Meanwhile, the author does not prove the proposed model is more interpretable. In Figure 5, the author claimed that the proposed model is interpretable because coefficient magnitude mass is concentrated on the very top of features, while the TF-IDF masses are distributed uniformly and are hard to interpret. However, we can find that the most positive high-level feature is hypertensive chronic kidney disease, and this finding is also obvious in the TF-IDF model. We can find that ESRD (end stage renal disease), dialysis, and hemodialysis are also on the top of the TF-IDF mass distribution. If we directly use TF-IDF to analyze the data, we can also obtain the conclusion that chronic kidney disease is the most important factor in readmission prediction. Meanwhile, the proposed model claims that the coronary atherosclerotic is a protective factor of readmission, which is unintuitive, but the author does not explain it. Therefore, it seems not inappropriate to claim that the TF-IDF is hard to interpret and the proposed model is more interpretable.


**Reproducibility:**

3: Could reproduce the results with some difficulty. The settings of parameters are underspecified or subjectively determined; the training/evaluation data are not widely available.

**Reviewer Confidence:**

4: Quite sure. I tried to check the important points carefully. It's unlikely, though conceivable, that I missed something that should affect my ratings.

---

> ### Author Rebuttal · Authors · 2023-08-28
>
> Though it is true that TF-IDF and BERT do perform better than our proposed approach at task prediction, it is worth reiterating that (as the reviewer notes) our primary objective is to introduce a new hybrid method that capitalizes on the zero-shot strengths of LLMs and the interpretability of linear models. We are therefore primarily interested in realizing interpretability in a flexible way,  rather than strictly maximizing predictive performance. In fact, the only trained portion of our model is a simple linear model over a very small number of features; it is not surprising that this fares worse than models defined over TF-IDF and BERT representations (which comprise >300x and >30,000x as many features respectively). However, the argument is that the proposed approach permits clinicians to specify arbitrary high-level features and then define a simple model over these; this is not possible with TF-IDF and BERT approaches. Furthermore, even putting aside the issue of allowing domain experts to craft novel features, these representations afford comparatively poor interpretability.
>
> In particular, regarding the reviewer’s interpretability concerns, we do not dispute that TF-IDF representations do contain individual features that are interpretable (e.g., compared to BERT or other dense neural representations) and our approach has some that are not (because feature extraction is not perfect). However, we argue that our features are interpretable more often than not by showing that learned features do align with the intuitions of clinicians before even seeing the model results: Table 2 shows most AUCs are much greater than 0.5. Figure 5 also demonstrates that TF-IDF one-word features are much harder to interpret than the hand-crafted features that our approach is capable of producing. TF-IDF, naively implemented, defines thousands of features; learning coefficients for each complicates any interpretation of model output.
>
> Per the reviewer’s suggestion, we will revise Figure 5 to explicitly mention the uninterpretable feature and explain that this is a product of the imperfect feature extraction because the ground truth features do not show this feature at the top.

---

### Official Review · Reviewer_DqbU · 2023-08-03

**Soundness:** 4

**Excitement:**

3: Ambivalent: It has merits (e.g., it reports state-of-the-art results, the idea is nice), but there are key weaknesses (e.g., it describes incremental work), and it can significantly benefit from another round of revision. However, I won't object to accepting it if my co-reviewers champion it.

**Paper Topic And Main Contributions:**

In this paper, the authors utilize the power of LLM for feature extraction in the medical domain. Plus, they investigate how the alignment of simple models' weights affects the prediction. They conduct experiments on medical datasets and provide discussion based on the results.

**Reasons To Accept:**

1, The task is an interesting task, especially in the medical domain.
2, They provide comprehensive experiment results and discussion.
3, This paper is well-written and easy to follow.

**Reasons To Reject:**

1, I do concern about the novelty of this work. To my best knowledge, most of the methods in this work are existing techniques.
2, I am wondering if this approach would raise the privacy issue. In the real-world setting, sensitive data would be strictly constrained to be fed into LLMs.

**Reproducibility:**

3: Could reproduce the results with some difficulty. The settings of parameters are underspecified or subjectively determined; the training/evaluation data are not widely available.

**Reviewer Confidence:**

3: Pretty sure, but there's a chance I missed something. Although I have a good feel for this area in general, I did not carefully check the paper's details, e.g., the math, experimental design, or novelty.

---

> ### Author Rebuttal · Authors · 2023-08-28
>
> 1. Though LLMs and linear models are existing techniques, to our knowledge, no prior work has used LLMs to extract binary or continuous features to be used as input for linear models. This is the core contribution on offer in this work. It is also novel that these features are specified by clinicians in natural language and extracted completely zero-shot.
>
> 2. With regards to concerns about privacy, we specifically choose FLAN-T5 as the LLM; variants of this model can be run locally within hospitals in a way that fully complies with privacy regulations and policy. (Indeed, we ran a variant at a hospital for this work.) We agree that one should not feed such data to APIs (e.g., OpenAI), but that is not what we are proposing here.

---

### Official Review · Reviewer_y8JT · 2023-08-04

**Soundness:** 3

**Excitement:**

3: Ambivalent: It has merits (e.g., it reports state-of-the-art results, the idea is nice), but there are key weaknesses (e.g., it describes incremental work), and it can significantly benefit from another round of revision. However, I won't object to accepting it if my co-reviewers champion it.

**Paper Topic And Main Contributions:**

This paper proposes to use LLM to extract information from the raw patient notes, which could serve as the interpretable features for training a simple classifier. This kind of method could achieve compatible results with other methods, while maintaining the interpretability which might be useful for real applications.

**Reasons To Accept:**

1. The idea to use LLM to generate examples for training is somewhat interesting;
2. The experiments are comprehensive and illustrate the effectiveness of this idea.

**Reasons To Reject:**

1. I may doubt the necessity of the linear classifier. What if we just let the LLM give the prediction according to the classified results from the templates? Maybe we could give the model several examples to see the few-shot classification performance;
2. From Table2, it seems that the large language models are not so good at prediction these diseases. AUC > 0.5 does not seem to be a good guarantee that the returned features are reasonable.

**Reproducibility:**

3: Could reproduce the results with some difficulty. The settings of parameters are underspecified or subjectively determined; the training/evaluation data are not widely available.

**Reviewer Confidence:**

3: Pretty sure, but there's a chance I missed something. Although I have a good feel for this area in general, I did not carefully check the paper's details, e.g., the math, experimental design, or novelty.

---

> ### Author Rebuttal · Authors · 2023-08-28
>
> 1. The core contribution here is the proposal to use LLMs to extract high-level, interpretable features from EHR to be used as features in a simple linear model that relates these to a “downstream” label of interest (e.g., 30-day readmission). This is to permit interpretability: Once trained, the model coefficients (each associated with a single high-level feature) can be inspected, on a global and per-instance level. Therefore, the primary purpose of the linear model is to relate high-level features to the output variable of interest. As a point of comparison, we do evaluate using LLMs to directly perform the “downstream” inference (Figure 4, black bars). But note that this is entirely blackbox. We are not *primarily* interested in the classification performance in and of itself; rather, we are proposing a new hybrid method that combines strengths of LLMs with those of linear models. We will improve the presentation of the work to further clarify the contribution and aims.
>
> 2. Table 2 does not report prediction performance. Rather it shows, for each task, how well the top features of the model align with expert intuition (i.e., the features that a clinician selected before seeing any model results or examples that should most contribute to a prediction), and it is impressive because it is completely unsupervised and purely auxiliary to the task on which the model was trained. The fact that this is clearly better than random is meaningful. We will revise the text to make sure this is communicated clearly in future versions of the work.

---

### Meta-Review · Area_Chair_KJKA · 2023-09-15

**Recommendation:** 3

**Metareview:**

The paper proposes CHiLL (Crafting High-Level Latents), an approach for natural-language specification of features for linear models. The authors conduct experiments on medical datasets and discuss the impact of weight alignment in simple models.

Reviewers appreciate the interesting task addressed in the paper and commend the comprehensive experiments and clear writing style. They find the experiments illustrative and the task relevant, especially in the medical domain. However, concerns regarding the novelty of the work, privacy implications, and the necessity of the linear classifier are raised.

In light of the reviewers' comments, the paper's contributions are evaluated. The paper's strengths lie in its attempt to bridge the gap between LLMs and interpretable features in the medical domain. However, the lack of clear evidence demonstrating the superiority of the proposed approach over traditional methods like TF-IDF and the absence of explanations for certain model decisions weaken the paper's overall impact.

Despite concerns raised by reviewers, the paper receives an average score above the acceptance board-line, indicating its potential value. Therefore, considering the overall positive sentiment and the potential significance of the paper's contributions, the decision should lean toward acceptance. However, the authors are encouraged to address the reviewers' suggestions, particularly by providing more robust evidence of the proposed method's superiority and further clarifying the interpretability aspect.

---

### Decision · Program_Chairs · 2023-10-07

**Decision:**

Accept-Findings

**Comment:**

The paper proposes CHiLL (Crafting High-Level Latents), an approach for natural-language specification of features for linear models. The authors conduct experiments on medical datasets and discuss the impact of weight alignment in simple models.

Reviewers appreciate the interesting task addressed in the paper and commend the comprehensive experiments and clear writing style. They find the experiments illustrative and the task relevant, especially in the medical domain. However, concerns regarding the novelty of the work, privacy implications, and the necessity of the linear classifier are raised.

In light of the reviewers' comments, the paper's contributions are evaluated. The paper's strengths lie in its attempt to bridge the gap between LLMs and interpretable features in the medical domain. However, the lack of clear evidence demonstrating the superiority of the proposed approach over traditional methods like TF-IDF and the absence of explanations for certain model decisions weaken the paper's overall impact.

Despite concerns raised by reviewers, the paper receives an average score above the acceptance board-line, indicating its potential value. Therefore, considering the overall positive sentiment and the potential significance of the paper's contributions, the decision should lean toward acceptance. However, the authors are encouraged to address the reviewers' suggestions, particularly by providing more robust evidence of the proposed method's superiority and further clarifying the interpretability aspect.